# Bisphenol AF Promoted the Growth of Uterus and Activated Estrogen Signaling Related Targets in Various Tissues of Nude Mice with SK-BR-3 Xenograft Tumor

**DOI:** 10.3390/ijerph192315743

**Published:** 2022-11-26

**Authors:** Mengjie Yu, Qianqian Tang, Bingli Lei, Yingxin Yang, Lanbing Xu

**Affiliations:** Institute of Environmental Pollution and Health, School of Environmental and Chemical Engineering, Shanghai University, Shanghai 200444, China

**Keywords:** bisphenol AF, Balb/cA nude mice with SK-BR-3 xenograft tumor, estrogenic activity, organ coefficient, G protein-coupled estrogen receptor 1

## Abstract

Environmental estrogens can promote the growth, migration, and invasion of breast cancer. However, few studies evaluate adverse health impacts of environmental estrogens on other organs of breast cancer patients. Therefore, the present study investigated the effects of environmental estrogen bisphenol AF (BPAF) on the main organs of female Balb/cA nude mice with SK-BR-3 xenograft tumor by detecting the organ development and gene expression of targets associated with G protein-coupled estrogen receptor 1 (GPER1)-mediated phosphatidylinositol 3-kinase/protein kinase B (PI3K/Akt) and mitogen-activated protein kinase (MAPK) signaling pathways in hypothalamus, ovary, uterus, liver, and kidney. The results showed that BPAF at 20 mg/kg bw/day markedly increased the uterine weight and the uterine coefficient of nude mice compared to SK-BR-3 bearing tumor control, indicating that BPAF promoted the growth of uterus due to its estrogenic activity. Additionally, BPAF significantly up-regulated the mRNA relative expression of most targets related to nuclear estrogen receptor alpha (ERα) and GPER1-mediated signaling pathways in the hypothalamus, followed by the ovary and uterus, and the least in the liver and kidney, indicating that BPAF activated different estrogen activity related targets in different tissues. In addition, BPAF markedly up-regulated the mRNA expression of GPER1 in all tested tissues, and the molecular docking showed that BPAF could dock into GPER1. Because gene change is an early event of toxicity response, these findings suggested that BPAF might aggravate the condition of breast cancer patients through exerting its estrogenic activity via the GPER1 pathway in various organs.

## 1. Introduction

Bisphenol AF (BPAF), as an alternative substitute of bisphenol A (BPA), has been widely applied to produce a variety of everyday household items and its global usage is estimated to increase in the next few years [1]. BPAF had been found in different environmental medium [2,3,4] with detection rate similar to BPA [5] and exists extensive exposure to human body by different pathways [6,7]. Additionally, BPAF can be transmitted to infants through breast milk [8]. Increasing evidence suggested that BPAF have estrogenic activity similar to or greater than that of BPA [9,10,11]. It was considered as an environmental endocrine disruptor (EED) due to its estrogen activity [12].

Breast carcinoma is the third most common cancer in the world, and it is the mostly diagnosed tumor in women [13]. A large number of epidemiological investigation results indicated that the continuous increase of endogenous estrogen level could significantly increase the occurrence and progression of breast cancer and environmental estrogens played an important role in this process [14,15,16,17]. Multiple genes are involved in the complicated process of breast carcinoma [18]. Estrogen receptor alpha/beta (ERα/β) is classical nuclear estrogen receptors and G protein-coupled estrogen receptor 1 (GPER1) is a novel member estrogen receptor, which are widely distributed in various tissues and organs of the human body [19]. Approximately 75% of breast tumors showed the positive expression of ERα and approximately 50% of breast cancer showed positive expression of GPER1. The in vitro study also showed that environmental estrogens increased the proliferation and transformation of ERα-positive breast tumor cells [20]. For in vivo research, the nude mouse xenograft tumor model has become an important tool to study tumor growth, metastasis, and pathological morphology because the genes of nude mice are almost the same as that of human [21]. Several studies found that environmental estrogens promoted the growth and development breast of breast cancer xenograft tumor in nude mice by activating ERα or GPER1 [22,23]. However, few studies are available about the effects of environmental estrogens on the other tissues of nude mice with breast cancer xenograft tumor. It is very important to perform this research for evaluating whether environmental estrogens can aggravate the condition of breast cancer patients by exerting estrogenic activity in other main organs.

Our previous study observed that BPAF up-regulated the mRNA levels of most targets associated with GPER1-mediated phosphatidylinositol 3-kinase/protein kinase B (PI3K/Akt) and mitogen-activated protein kinase (MAPK) signaling pathways in ER-negative human breast cancer SK-BR-3 cells and induced the cell viability [24]. In vivo study demonstrated that BPAF increased the weight of SK-BR-3 xenograft tumor in nude mice and up-regulated the mRNA levels of most targets related to GPER1-mediated MAPK and PI3K/Akt signaling pathways in SK-BR-3 tumor tissue [24]. However, after BPAF enter the organism, BPAF can reach various organs of the body through transportation and metabolism, whether BPAF can activate the expression of target genes related to estrogen signaling pathway in main organs and affect the development of these organs in nude mice with SK-BR-3 xenograft tumor is unclear.

Therefore, in this study, GPER1-mediated MAPK and PI3K/Akt signaling pathways were still used as main signal targets. The gene expression of 19 targets including *GPER1*, estrogen receptor alpha (*ESR1*), epidermal growth factor receptor (*EGFR*), *MAPKs*, *Akt*, and their downstream targets associated with estrogen signaling pathway in the main organs (hypothalamus, ovary, uterus, liver, kidney) of nude mice with SK-BR-3 xenograft tumor were analyzed by RT-qPCR technology. Additionally, the molecular docking was used to study the interaction between BPAF and GPER1. We also investigated the effects of BPAF on the development of various organs of nude mice by calculating organ coefficients. This study will highlight the hazards of BPAF on breast cancer patients due to its estrogenic activity.

## 2. Materials and Methods

### 2.1. Chemicals

Bisphenol AF (BPAF, Cas No. 1478-61-1) was purchased from Tokyo Chemical Industry (TCI, Tokyo, Japan) and its purity is higher than 98%. The stock solution of BPAF dissolved in dimethyl sulfoxide (DMSO, 99.5%, AMRESCO, Radnor, PA, USA) was kept at −20 °C. The source and maintenance of SK-BR-3 cells can be found in our previous study [24].

### 2.2. In Vivo Xenograft Animal Model Experiment

The details of experimental design could be found in our previous study [24]. Briefly, 24 female *BALB/c* nude mice were purchased from Shanghai Jihui Experimental Animal Feeding Co., Ltd. (Shanghai, China). These mice were maintained under a 12 h light: 12 h dark cycle and had a normal diet and free access to tap water (Suzhou Xishan Zhongke Pharmaceutical Research and Development Co., Ltd., Suzhou, China). The SK-BR-3 human breast cancer cells in the logarithmic growth phase were suspended in serum-free medium and mixed with BD Matrigel matrix gel (BD company, Franklin Lakes, NJ, USA) in a ratio of 1:1 (*v:v*). Then, 18 of 24 female *Balb/cA* nude mice that had been adaptively raised for one week were used to make SK-BR-3 breast cancer xenograft animal model by inoculating with SK-BR-3 cell suspension in the right armpit of the nude mice at 3 × 10^7^ cells/mice. SK-BR-3 tumors could be observed clearly in the right axilla of nude mice approximately 10 days after inoculation and tumor volume (TV) is about 64.63 ± 28.57 (mm^3^) which was calculated according to the formula (TV = a × (b^2^)/2). The long diameter (a) and short diameter (b) of the tumors were measured with vernier caliper [24]. Then, 18 nude mice models with SK-BR-3 xenograft tumor were divided into three groups with 6 nude mice in each group. These three groups were SK-BR-3 bearing tumor control, 20 mg/kg bw/day BPAF treatment and 100 mg/kg bw/day BPAF treatment, respectively. The other 6 nude mice without SK-BR-3 xenograft tumor were set as normal control. The dosages of the two BPAF exposure groups were set according to the oral half lethal dose (LD_50_) of 3400 mg/kg in rats exposed acutely to BPAF [25]. Based on the weight of the nude mice, BPAF solutions dissolved in corn oil was orally administered to nude mice with SK-BR-3 xenograft tumor for 15 days in dosages of 20 and 100 mg/kg bw/day, which were approximately equivalent to 1/150 and 1/30 LD_50_, respectively. SK-BR-3 bearing tumor control mice and normal control mice received only corn oil treatment.

During BPAF administration, the mental state, skin colour, and diet of the nude mice were observed everyday and there was no significant change compared to the control groups. The weight of mice was recorded once a day at a fixed time point. Mice were sacrificed with a cervical vertebrae luxation method at the end of the fifteenth day to collect hypothalamus, ovary, uterus, liver, and kidney tissues for further organ coefficient calculation and gene expression analysis.

### 2.3. Organ Coefficient

The organ coefficients were calculated by comparing the organ weight to the whole-body weight of nude mice. For example, liver coefficient = liver weight/body weight of nude mice × 100% [26].

### 2.4. RT-qPCR

Tissue RNA was isolated using buffer RL lysate. Qualified RNA was employed to reversely transcribe into complementary deoxyribose nucleic acid (cDNA) using the reverse transcription kit (ReverTra Ace^®^, TOYOBO. Co., Ltd., Japan). Real-time PCR Master Mix SYBR Green^®^ (TOYOBO. Co., Ltd., Osaka, Japan) was employed for RT-qPCR with cDNA as the template, which was performed using the Bio-Rad CFX 96 PCR system (Bio-Rad, Hercules, CA, USA). Relative gene expression was normalized to *β-actin* and was calculated by 2^−ΔΔCt^ [25]. The forward and reverse primers of target genes were the same as the sequences described in our previous study [24].

### 2.5. Data Analysis

The results were characterized as mean ± standard deviation (SD). The difference between BPAF treatment group and the control group was examined using a one-way ANOVA test followed by the least-significant difference (LSD) multiple range test using SPSS 19.0 (Chicago, IL, USA). Differences were considered to be statistically significant when * *p* < 0.05.

## 3. Results

### 3.1. BPAF Increased Organ Coefficient of Uterus of Nude Mice

Our previous study found that BPAF at 20 and 100 mg/kg bw/day markedly increased the relative tumor volume (RTV) and weight of SK-BR-3 xenograft tumor of nude mice relative to SK-BR-3 bearing tumor control [24]. In this study, we further studied the impacts of BPAF treatment on body weight and various organ coefficients of nude mice with SK-BR-3 xenograft tumor. It was found that BPAF treatment for 15 days had no significant influence on the weight of nude mice with SK-BR-3 tumor relative to SK-BR-3 bearing tumor control or normal control (Figure 1A). Appendix A showed the change of body weight of nude mice with the increase of BPAF treatment days. However, BPAF at 20 mg/bw kg/day significantly increased the weight of uterus (*p* = 0.016) and ovary (*p* = 0.04) of nude mice relative to SK-BR-3 bearing tumor control (Figure 1A). In addition, one significant increase in the uterine coefficient of nude mice treated with 20 mg/bw kg/day BPAF was also observed relative to SK-BR-3 bearing tumor control (*p* = 0.018) or normal control (*p* = 0.011) (Figure 1B). Although BPAF at 20 mg/bw kg/day increased the value of ovarian coefficient, no significant difference between BPAF treatment and SK-BR-3 bearing tumor control was observed (*p* = 0.054). Besides, BPAF treatments also had no significant influence on the weight of the other tested organs and organ coefficients of nude mice (Appendix A).

### 3.2. BPAF Elevated the Expression of Target Genes in Hypothalamus of Nude Mice

Figure 2 showed the effects of BPAF on the mRNA relative levels of related genes in hypothalamus of neuron system in nude mice with SK-BR-3 xenograft tumor. BPAF at 20 mg/kg bw/day significantly only up-regulated the gene expression of *Akt* and *TFF1* in hypothalamus, while the mRNA expression of the other 17 genes did not present a significant difference relative to SK-BR-3 bearing tumor control. At 100 mg/kg bw/day, BPAF induced a significant upregulation in the gene expression of 13 targets and the mRNA levels of the other six genes had no significant difference relative to SK-BR-3 bearing tumor control.

### 3.3. BPAF Elevated the mRNA Relative Levels of Genes in Reproductive Organs of Nude Mice

The effects of BPAF on the mRNA relative levels of related genes in reproductive organs (ovary and uterus) of nude mice with SK-BR-3 xenograft tumor are shown in Figure 3A–D. For ovarian tissue, compared with SK-BR-3 bearing tumor control, the mRNA level of only *MAPK3* in ovarian tissue was significantly up-regulated when the exposure dose of BPAF was 20 mg/kg bw/day, and the gene expression of five targets (*MAPK1*, *TFF1*, *Fos*, *FOXO1*, *CCND1*) were significantly down-regulated (Figure 3A). The mRNA relative levels of the other 11 genes were not significantly changed. When the nude mice were treated with 100 mg/kg bw/day BPAF, the gene expression of 11 targets (*GPER1*, *ESR1*, *EGFR, MAPK6, JNK*, *Akt*, *Myc*, *EBP1*, *FKBP1A*, *PGR, CCND1*) were significantly up-regulated and the mRNA levels of *Fos* were significantly down-regulated, while the mRNA levels of the other seven genes had no the significant change relative to SK-BR-3 bearing tumor control. Regarding BPAF at 20 mg/kg bw/day, the data distribution of mRNA relative levels of 19 genes in ovary was very clustered, between 0.6 and 1.4, while BPAF at 100 mg/kg bw/day, it was relatively scattered, between 0.5 and 3.0 (Figure 3C).

For uterus, relative to SK-BR-3 bearing tumor control, BPAF at 20 mg/kg bw/day markedly up-regulated the gene expression *Akt* and *Myc* in uterine tissue, while the expression of other 17 genes had no a statistical difference (Figure 3B). When the nude mice were treated with 100 mg/kg bw/day BPAF, the mRNA expression of 11 genes (*GPER1*, *ESR1*, *EGFR, MAPK3*, *MAPK6*, *Akt*, *Myc*, *TFF1*, *PGR*, *SRF*, *CCND1*) in uterine tissue were significantly up-regulated, while the gene expression of the other eight targets did not present a significant change relative to SK-BR-3 bearing tumor control. BPAF at 20 and 100 mg/kg bw/day, the data distributions of the relative expression levels of 19 genes in uterus were between 0.8 and 1.5 and 0.8 and 3.0, respectively (Figure 3D). Similar to the ovary, with BPAF at 20 mg/kg bw/day, the data distribution of the mRNA relative levels of targets was very clustered, while with BPAF at 100 mg/kg bw/day, it was relatively scattered.

### 3.4. BPAF Elevated the mRNA Relative Levels of Targets in Metabolic Organs of Nude Mice

The influence of BPAF on the gene relative expression of related targets in metabolic organs (liver and kidney) of nude mice were shown in Figure 4A–D. For liver tissue, relative to SKBR3 bearing tumor control, BPAF at 20 mg/kg bw/day significantly up-regulated the gene expression of four genes (*MAPK3*, *MAPK6*, *EBP1*, *CCND1*) (Figure 4A), whereas the mRNA relative levels of the other 15 genes had no a statistical difference. When the nude mice were treated with 100 mg/kg bw/day BPAF, the mRNA levels of only *GPER1*, *Fos*, and *CCND1* were significantly up-regulated, while the gene expression of seven targets, *ESR1*, *MAPK1*, *MAPK6*, *JNK*, *Akt*, *EBP1*, and *FKBP1A* were significantly down-regulated and the gene expression of the other nine targets had no statistically significant difference relative to SK-BR-3 bearing tumor control. BPAF at 20 and 100 mg/kg bw/day, the data distributions of the mRNA relative levels of 19 genes in liver were mainly between 0.8 and 1.5 and 0.5 and 1.0, respectively (Figure 4C).

In kidney tissue, BPAF at 20 mg/kg bw/day only caused a significant upregulation of *EBP1* gene, the mRNA levels of the other 18 genes had no statistical difference relative to SK-BR-3 bearing tumor control (Figure 4B). At 100 mg/kg bw/day, BPAF resulted in the significant upregulation in gene expression of seven targets (*GPER1*, *JNK*, *Akt*, *Fos*, *FOXO1*, *PDZK1*, *SRF*), while the gene expression of the other 12 targets were not significantly changed compared to the SK-BR-3 bearing tumor control. With BPAF at 20 and 100 mg/kg bw/day, the data distribution of the mRNA relative levels of 19 genes in kidney were mainly between 0.8 and 1.3 and 0.9 and 1.5, respectively (Figure 4D).

## 4. Discussion

The establishment of nude mice xenograft tumor model is not only an important means for studying the impact of environment contaminants on the characterization and progression of human breast cancer [23,27], but also provides a good condition for investigating the effects of environmental estrogens on the growth and development of the other main organs of breast cancer patients by exerting their estrogenic activity. This is of great practical significance for studying the comprehensive effects of estrogen compounds on the condition of breast cancer patients. This study found that BPAF markedly up-regulated the mRNA relative levels of some target genes related to ERα and GPER1-mediated estrogenic signaling pathways in the hypothalamus, ovary, uterus, liver, and kidney and BPAF at 20 mg/kg bw/day markedly increased uterine weight and uterine coefficient of nude mice with SK-BR-3 xenograft tumor. The uterine weight gain of mice is an important index to judge the estrogenic activity of compounds [28,29]. The results indicated that BPAF affected the development of uterus by exerting its estrogenic activity in nude mice with SK-BR-3 tumor. In female mice treated neonatally with BPA, severe pathologies of the uterus including adenomyosis, leiomyomas, atypical hyperplasia, stromal polyps were observed [30]. In addition, BPA exposure showed fewer primordial follicles and increased growing follicles [31]. These severe pathological conditions are associated with an increased proliferation rate likely mediated by an estrogenic pathway [31,32]. Environmental estrogens can disrupt endocrine function by competing with endogenous estrogen to bind to estrogen nuclear receptor ERα/β [20,33], which is one of the most important causes of estrogen sensitive cancer, such as breast cancer [14]. Additionally, several studies have shown that environmental estrogens can also exert estrogenic activity by rapidly activating the endoplasmic reticulum independent signaling pathway mediated by membrane receptor GPER1 in a non-genomic way [34,35,36,37], while GPER1 activation depends on EGFR activation [38]. The present study found that BPAF also up-regulated the expression of ERα, GPER1, and EGFR in the hypothalamus, ovary, and uterus of nude mice with SK-BR-3 tumor.

GPER1 activation can mediate the rapid activation of multiple signaling pathways. PI3K/Akt and MAPK are two classical signaling pathways mediated by GPER1. MAPK includes Erk, JNK and p38MAPK family members, which is a common signal transduction element in a variety of signal transduction pathways [19]. PI3K is a cytosolic phosphatidylinositol kinase and Akt is its downstream protein kinase, which are closely related to the growth and survival of cells [39]. Several studies have reported that estrogens activated MAPK and PI3K/Akt signaling pathways by GPER1 which further regulated their downstream targets, such as EBP1, Elk-1, c-Fos, Myc, c-Jun, FOXO1, and CyclinD1, and participated in a variety of biological responses [40,41,42,43]. The present study also observed that BPAF significantly elevated the mRNA levels of *GPER1* and the gene expression of some targets associated with PI3K/Akt and MAPK signaling pathways in the hypothalamus, ovary, uterus, liver, and kidney of nude mice with SK-BR-3 xenograft tumor. In addition, BPAF at 20 mg/kg bw/day markedly increased the weight of ovary and uterus as well as the uterine coefficient of nude mice. These findings suggested that BPAF might activate MAPK and PI3K/Akt signaling pathways by GPER1 in different tissues of nude mice and affect the development of the main organs.

In addition, although BPAF up-regulated the mRNA levels of target genes related to PI3K/Akt and MAPK pathways, there existed differences between cells and tissues, as shown in Table 1. We observed that BPAF at 0.01 μM markedly up-regulated the gene mRNA levels of all tested 18 targets related to PI3K/Akt and MAPK signaling pathways in SK-BR-3 cells. In SK-BR-3 tumor tissue from nude mice, BPAF at 100 mg/kg bw/day significantly elevated the gene expression of 14 targets related to PI3K/Akt and MAPK signaling pathways. The present study found that BPAF at 100 mg/kg bw/day markedly up-regulated the gene expression of 13 targets related to PI3K/Akt and MAPK pathways in hypothalamus of nude mice with SK-BR-3 xenograft tumor. The mRNA levels of 11 target genes in the ovary and uterus were significantly up-regulated in nude mice treated with 100 mg/kg bw/day BPAF. In the liver and kidney, only several mRNA levels of target genes were up-regulated significantly. The SK-BR-3 cells are human breast cancer cells with GPER1-positive expression, which are very sensitive to estrogen chemicals [24]. Therefore, BPAF significantly up-regulated the gene expression of most targets associated with estrogen signaling pathways in SK-BR-3 cells or SK-BR-3 tumor tissue, indicating the estrogenic activity of BPAF again, which was also consistent with other in vitro studies concerning the estrogenic activity of BPAF [10,12].

In different tissues, BPAF-up-regulated target genes were also different. This may be related to the different functions of organs. By comparison, it was found that after BPAF treatment, the gene expression of most targets related to estrogen signaling pathway in hypothalamus was significantly up-regulated, followed by the ovary and uterus, and the least in liver and kidney. The nervous organ hypothalamus, as a main connection of between the central nervous system and the endocrine system, plays a key role in mediating physiological activity and homeostatic mechanism [44]. The hypothalamus can secrete various hormones and is very sensitive to exogenous hormones [45,46,47]. Ovaries and uterus, as main reproductive organs, are target organs of estrogen action and are also very sensitive to estrogen or environmental estrogens [29,48]. The liver and kidney are metabolic organs [49], which can metabolize environmental estrogens into substances with lower or stronger estrogen activity [50,51]. Additionally, it was reported that estrogen inactivation and excretion in women liver could be a defense mechanism against toxicities of estrogens [52]. Therefore, it was possible that BPAF showed a weaker estrogenic effect on the gene expression of targets associated with estrogenic signaling pathway in liver and kidney compared to other organs for above the reasons. These findings indicated that BPAF could activate different signal targets related to GPER1-mediated PI3K/Akt and MAPK signaling pathways in different tissues. In addition, from Table 1, we observed that BPAF could markedly up-regulate the gene expression of *GPER1* in SK-BR-3 cells and all tested tissues relative to the SK-BR-3 bearing tumor control. The molecular docking analysis showed that BPAF also had an interaction with GPER1 by forming hydrogen bonds with Cys205 and Cys207 that were similar to BPA (Figure 5). These results suggested that BPAF might accelerate the progression of breast cancer by directly promoting the growth of SK-BR-3 tumor and worsen the condition of breast cancer patients by indirectly affecting other main organs, which could be related to the activation of a novel estrogen member receptor GPER1 in a non-genomic way.

## 5. Conclusions

BPAF significantly elevated the mRNA levels of different target genes related to estrogen signaling pathways in different organs, including the hypothalamus, ovary, uterus, liver, and kidney of nude mice with Sk-BR-3 xenograft tumor. However, BPAF markedly up-regulated the mRNA levels of *GPER1* in all test organs. In addition, BPAF at 20 mg/kg bw/day markedly increased the uterine weight and uterine coefficient of nude mice. The findings indicated that BPAF exerted the estrogenic activity by activating the GPER1-mediated estrogen signaling pathway in different organs. Besides, the estrogenic activity of BPAF might aggravate the condition of breast cancer patients by adversely affecting the main organ development of breast cancer patients. The study provided a novel perspective to evaluate the adverse health impacts of BPAF on breast cancer patients with GPER1-positive expression.

## Figures and Tables

**Figure 1 ijerph-19-15743-f001:**
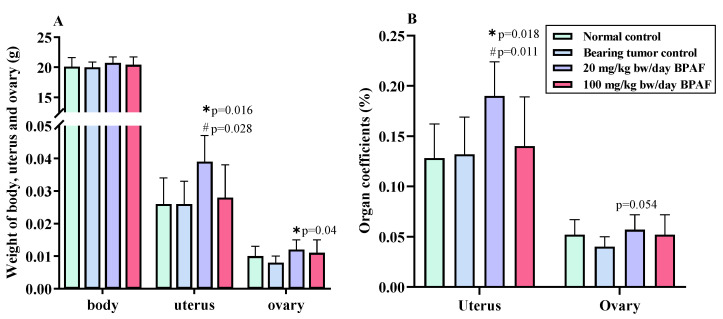
Effect of BPAF on the weight of body, uterus and ovary and organ coefficients of *Balb/cA* nude mice with SK-BR-3 xenograft tumor. (**A**): the wight of body, uterus and ovary; (**B**): Uterine and ovarian coefficients. The data represented the mean ± SD of six parallel mice samples. * *p* < 0.05, relative to SK-BR-3 bearing tumor control; ^#^
*p* < 0.05, relative to normal control.

**Figure 2 ijerph-19-15743-f002:**
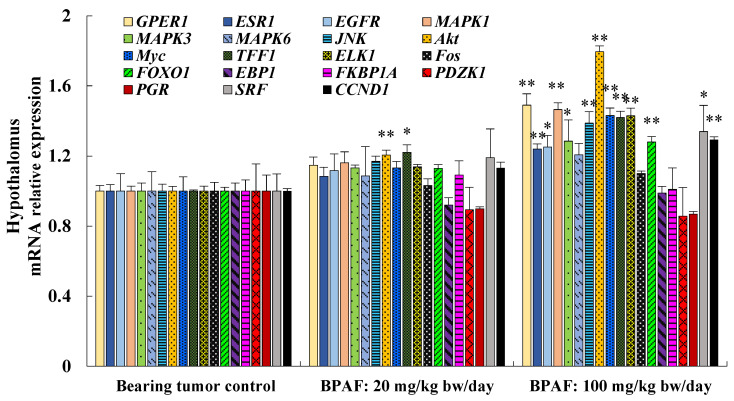
Effect of BPAF exposure for 15 days on the mRNA relative expression of 19 genes associated with the PI3K/Akt and MAPK pathways in hypothalamus of female nude mice with SK-BR-3 xenograft tumor. The data showed the mean ± SD of six parallel mice samples. * *p* < 0.05 and ** *p* < 0.01, relative to SK-BR-3 bearing tumor control. Note: GPER1: G protein-coupled estrogen receptor 1; ESR1: estrogen receptor alpha; EGFR: epidermal growth factor receptor; MAPK: mitogen-activated protein kinase; JNK: c-Jun N-terminal kinase; Akt: protein kinase B; Myc: myc proto-oncogene; TFF1: trefoil factor 1; ELK1: ETS domain-containing protein; Fos: fos proto-oncogene; FOXO1: forkhead box O1; EBP1: ErbB-3 binding protein 1; FKBP1A: FK-506 binding protein 1A; PDZK1: PDZ domain protein kidney 1; PGR: progesterone receptor; SRF: serum response factor; CCND1: cyclin D1.

**Figure 3 ijerph-19-15743-f003:**
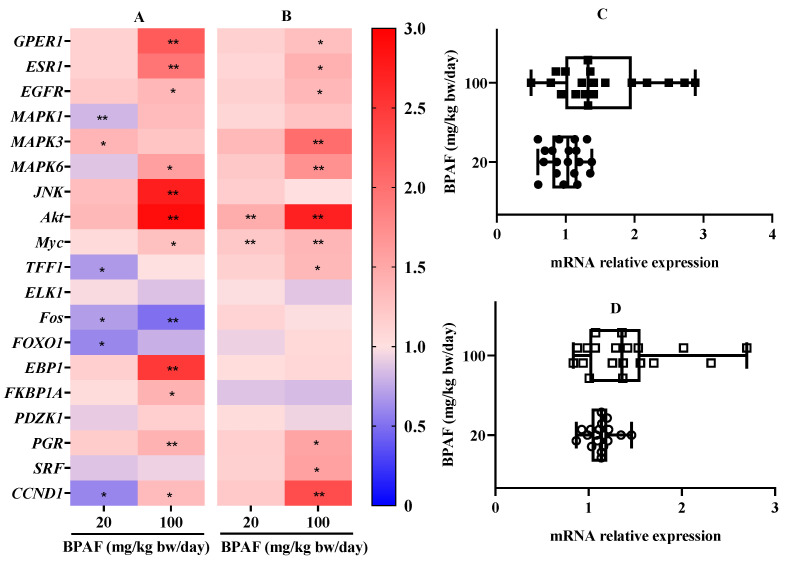
Effect of BPAF exposure for 15 days on the mRNA relative expression of 19 genes associated with the PI3K/Akt and MAPK pathways in in ovary and uterus of female nude mice with SK-BR-3 xenograft tumor. (**A**): the mRNA relative expression of 19 genes in ovary; (**B**): the mRNA relative expression of 19 genes in uterus; (**C**): the data distribution of mRNA relative levels of 19 genes in ovary; (**D**): the data distribution of mRNA relative levels of 19 genes in uterus. The data represented the mean of six parallel mice samples. * *p* < 0.05 and ** *p* < 0.01, relative to SK-BR-3 bearing tumor control.

**Figure 4 ijerph-19-15743-f004:**
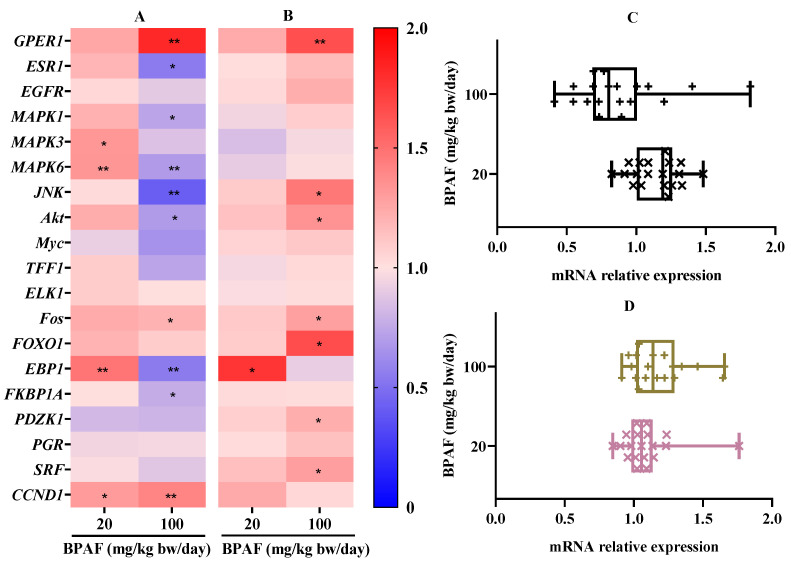
Effect of BPAF exposure for 15 days on the mRNA relative levels of 19 genes associated with the PI3K/Akt and MAPK pathways in liver and kidney of female nude mice with SK-BR-3 xenograft tumor. (**A**): the mRNA relative expression of 19 genes in liver; (**B**): the mRNA relative expression of 19 genes in kidney; (**C**): the data distribution of mRNA relative levels of 19 genes in liver; (**D**): the data distribution of mRNA relative levels of 19 genes in kidney. The data represented the mean of six parallel mice samples. * *p* < 0.05 and ** *p* < 0.01, relative to SK-BR-3 bearing tumor control.

**Figure 5 ijerph-19-15743-f005:**
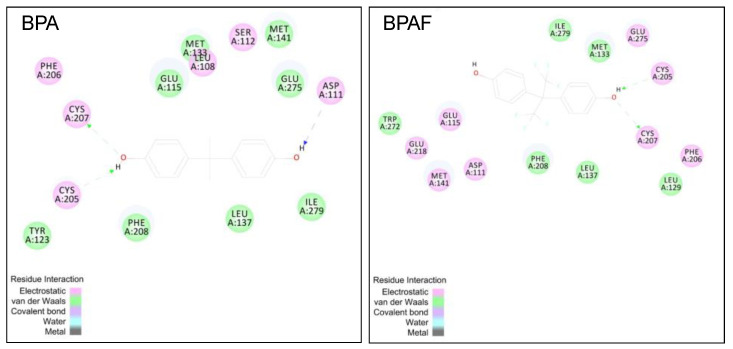
Interaction between BPAF and GPER1. The Discovery Studio 4.0 software package was used to dock BPAF and BPA to GPER1 (Accelrys Software Inc., San Diego, CA, USA). The Cdocker protocol was used to perform the protocol. In this Cdocker protocol, 20 random ligand conformations were generated and refined through grid-based simulated annealing in the binding site of GPER1, and the top ranked postures from the docking were chosen [53].

**Table 1 ijerph-19-15743-t001:** The summary of mRNA levels of tested genes associated with GPER1-mediated PI3K/Akt and MAPK signaling pathways in SK-BR-3 cells or in various tissues from nude mice with SK-BR-3 xenograft tumor.

Group	SK-BR-3 Cells ^a^	SK-BR-3 Tumor ^a^		Hypothalamus ^b^		Ovary ^b^		Uterus ^b^		Liver ^b^		Kidney ^b^	
BPAF	μM	mg/kg bw/day											
	0.01	20	100	20	100	20	100	20	100	20	100	20	100
*GPER1*	↑ **	→	↑ **	→	↑ **	→	↑ **	→	↑ *	→	↑ **	→	↑ **
*ESR1*	-	-	-	→	↑ **	→	↑ **	→	↑ *	→	↓ *	→	→
*EGFR*	↑ **	→	↑ *	→	↑ *	→	↑ *	→	↑ *	→	→	→	→
*MAPK1*	↑ **	→	↑ *	→	↑ **	↓ **	→	→	→	→	↓ *	→	→
*MAPK3*	↑ **	→	↑ **	→	↑ *	↑ *	→	→	↑ **	↑ *	→	→	→
*MAPK6*	↑ **	→	↑ **	→	→	→	↑ *	→	↑ **	↑ **	↓ **	→	→
*JNK*	↑ **	↑ *	↑ **	→	↑ **	→	↑ **	→	→	→	↓ **	→	↑ *
*Akt*	↑ **	→	↑ *	↑ **	↑ **	→	↑ **	↑**	↑ **	→	↓ *	→	↑ *
*Myc*	↑ **	↑ **	↑ **	→	↑ **	→	↑ *	↑**	↑ **	→	→	→	→
*TFF1*	↑ **	→	↑ *	↑ *	↑ **	↓ *	→	→	↑ *	→	→	→	→
*ELK1*	↑ **	→	↑ *	→	↑ **	→	→	→	→	→	→	→	→
*Fos*	↑ **	↑ **	↑ **	→	→	↓ *	↓ **	→	→	→	↑ *	→	↑ *
*FOXO1*	↑ **	↓ **	↓ **	→	↑ **	↓ *	→	→	→	→	→	→	↑ **
*EBP1*	↑ **	↓ **	→	→	→	→	↑ **	→	→	↑ **	↓ **	↑ *	→
*FKBP1A*	↑ **	→	↑ **	→	→	→	↑ *	→	→	→	↓ *	→	→
*PDZK1*	↑ **	↑ *	↑ **	→	→	→	→	→	→	→	→	→	↑ *
*PGR*	↑ **	→	↓ *	→	→	→	↑ **	→	↑ *	→	→	→	→
*SRF*	↑ **	↑ *	↑ *	→	↑ *	→	→	→	↑ *	→	→	→	↑ *
*CCND1*	↑ **	→	→	→	↑ **	↓ *	↑ *	→	↑ **	↑ *	↑ **	→	→

Note: ^a^ the data come from the previous study [24]; ^b^ the data come from this study; ↑ means up-regulation; ↓ means down-regulation; → means no significant difference; - means no detection. * *p* < 0.05; ** *p* < 0.01, compared with 0.1% DMSO or SK-BR-3 bearing tumor control.

## Data Availability

The data can be provided as required.

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
