# Peer review of "Bisphenol AF Promoted the Growth of Uterus and Activated Estrogen Signaling Related Targets in Various Tissues of Nude Mice with SK-BR-3 Xenograft Tumor"

_ijerph, 2022, doi:10.3390/ijerph192315743_

Round 1

Reviewer 1 Report

This study is the continuation of the evaluation of the molecular mechanism of BPAF-induced proliferation in SKBR-3 human breast cancer. It is a well design study with interesting conclusions which totally supported by the results. The manuscript can be accepted for publication after some minor modifications:

1.       Authors must explain the reason they chose the dosages of 20 and 100 mg/kg bw/day. Did they conduct any toxicity studies?

2.       They must also report any other side effects they noticed during drug administration except from weight loss (hair loss, anorexia, hypnalgia, etc.)

3.       In the xenograft tumor development, it is not scientifically correct to mention that the experimentation began when the tumor ‘’could be seen clearly’’. Authors must mention the exact volume of the tumor they measured with caliper.

Author Response

Reviewer1:

This study is the continuation of the evaluation of the molecular mechanism of BPAF-induced proliferation in SKBR-3 human breast cancer. It is a well design study with interesting conclusions which totally supported by the results. The manuscript can be accepted for publication after some minor modifications:

  1. Authors must explain the reason they chose the dosages of 20 and 100 mg/kg bw/day. Did they conduct any toxicity studies?

Reply: “BPAF solutions dissolved in corn oil were orally administered to the nude mice with SK-BR-3 xenograft tumor for 15 days.” was changed into “The dosages of the two BPAF exposure groups were set according to the oral half lethal dose (LD50) of 3400 mg/kg in rats exposed acutely to BPAF [25]. Based on the weight of the nude mice, BPAF solutions dissolved in corn oil was orally administered to nude mice with SK-BR-3 xenograft tumorfor 15 days in dosages of 20 and 100 mg/kg bw/day, which were approximately equivalent to 1/150 and 1/30 LD50, respectively” was added.

  1. They must also report any other side effects they noticed during drug administration except from weight loss (hair loss, anorexia, hypnalgia, etc.)

Reply: “During BPAF administration, the mental state, skin colour and diet of the nude mice were observed everyday and there was no significant change compared to the control groups. ” was added.

  1. In the xenograft tumor development, it is not scientifically correct to mention that the experimentation began when the tumor ‘’could be seen clearly’’. Authors must mention the exact volume of the tumor they measured with caliper.

Reply: “When SK-BR-3 xenograft tumor in the right armpit could be seen clearly.” was changed into “SK-BR-3 tumors could be observed clearly in the right axilla of nude mice approximately 10 days after inoculation and tumor volume (TV) is about 64.63±28.57 (mm3) which was calculated according to the formula (TV =a× (b2)/2). The long diameter (a) and short diameter (b) of the tumors were measured with vernier caliper ”

Reviewer 2 Report

The present manuscript is focused on the role of BPAF in BC cells and on the development of various organs of nude mice by calculating organ coefficients.

The manuscript is well written, well presented and the pictures are well performed.

I have few questions:

1. The authors should add mice photographs relative to their experiments.

2. Two isoforms of estrogen receptor

(ER), alpha (a) and beta (b), are expressed in normal and malignant ovarian cells.

Expression of ERb is higher than that of ERa not only in the normal

ovary but also in benign ovarian tumors (Enmark et al., 1997;

Kuiper and Gustafsson, 1997; Pujol et al., 1998), whereas the

opposite pattern characterizes development of malignant ovarian

cancer (Bardin et al., 2004). ERb seems to have a protective role,

inhibiting cell migration and proliferation, and inducing apoptosis

(Bardin et al., 2004). High levels of the two ER isoforms are present

in human ovarian epithelial cancer cells compared to normal hu-

man ovarian epithelial cells, confirming that ER expression is

probably associated with an unfavorable prognosis (Munstedt et al.,

2000). What about the changes in ERb expression after BPAF treatment? What about the ERa/ERb ratio?

3.  In aged female mice treated

neonatally with BPA, alterations

in estrus cyclicity, ovulation and ovarian morphology are observed.(Suzuki et al.,

2002). This is accomplished by a higher incidence of cystic ovaries, cystic endometrial hy-

hyperplasia, endometriosis and progressive proliferative lesions of

the oviduct, and other pathologies of the uterus including adeno-

myosis, leiomyomas, atypical hyperplasia and stromal polyps

(Newbold et al., 2007). These severe pathological conditions are

associated with an increased proliferation rate likely mediated by

an estrogenic pathway (Newbold et al., 2007; Rodriguez et al.,

2010). Did the authors observe similar findings with BPAF? (see. Mol Cell Endocrinol

  2017 Dec 5;457:35-42.

 doi: 10.1016/j.mce.2017.02.045).

Minor revisions:

please correct typos

Author Response

Reviewer 2:

The present manuscript is focused on the role of BPAF in BC cells and on the development of various organs of nude mice by calculating organ coefficients. The manuscript is well written, well presented and the pictures are well performed. I have few questions:

  1. The authors should add mice photographs relative to their experiments.

Reply: This is a picture of tumor formation in mice in the experiment. Since this experiment was completed in 2019, other experimental photos have not been saved.

  1. Two isoforms of estrogen receptor (ER), alpha (a) and beta (b), are expressed in normal and malignant ovarian cells. Expression of ERb is higher than that of ERa not only in the normal ovary but also in benign ovarian tumors (Enmark et al., 1997; Kuiper and Gustafsson, 1997; Pujol et al., 1998), whereas the opposite pattern characterizes development of malignant ovarian cancer (Bardin et al., 2004). ERb seems to have a protective role, inhibiting cell migration and proliferation, and inducing apoptosis (Bardin et al., 2004). High levels of the two ER isoforms are present in human ovarian epithelial cancer cells compared to normal human ovarian epithelial cells, confirming that ER expression is probably associated with an unfavorable prognosis (Munstedt et al., 2000). What about the changes in ERb expression after BPAF treatment? What about the ERa/ERb ratio?

Reply:  ERÉ‘ and ERβ are two classical estrogen nuclear receptors. Many studies mentioned in above literature have found that ERÉ‘ and ERβ played an important role in development of malignant ovarian cells. In this study, we used nude mice with SK-BR-3 xenograft tumor as animal model to study the estrogenic effects of BPAF on breast cancer patients. we mainly study the role of novel estrogen member receptor GPER1 in BPAF-induced estrogen activity because GPER1 presented positive expression and ERα presented negative expression in SKBR3 cells. Therefore, we did not detect the expression of ERβ in different tissues.

  1. In aged female mice treated neonatally with BPA, alterations in estrus cyclicity, ovulation and ovarian morphology are observed. (Suzuki et al., 2002). This is accomplished by a higher incidence of cystic ovaries, cystic endometrial hy-hyperplasia, endometriosis and progressive proliferative lesions of the oviduct, and other pathologies of the uterus including adeno-myosis, leiomyomas, atypical hyperplasia and stromal polyps (Newbold et al., 2007). These severe pathological conditions are associated with an increased proliferation rate likely mediated by an estrogenic pathway (Newbold et al., 2007; Rodriguez et al., 2010). Did the authors observe similar findings with BPAF? (see. Mol Cell Endocrinol 2017 Dec 5;457: 35-42. doi: 10.1016/j.mce.2017.02.045).

Reply: In this study, BPAF at 20 mg/bw kg/day significantly increased the weight of uterus (p=0.016) and ovary (p=0.04) of nude mice with SK-BR-3 xenograft tumor relative to SK-BR-3 bearing tumor control. Additionally, BPAF significantly up-regulated mRNA relative expression of most targets related to nuclear estrogen receptor alpha (ERα) and GPER1-mediated signaling pathways in hypothalamus, followed by in ovary and uterus, and the least in liver and kidney, indicating that BPAF activated different estrogen activity related targets in different tissues. The results showed that the uterine weight gain are also likely mediated by an estrogenic pathway which is similar with BPA.

However, in this study, we mainly used in vivo nude mice with SK-BR-3 xenograft tumor as animal model to study the estrogenic effects of BPAF on breast cancer patients. Because gene change is early event of toxicity response, these findings suggested that BPAF might aggravate the condition of breast cancer patients through exerting its estrogenic activity via GPER1 pathway in various organs.

The purpose is to discuss more deeply, “In female mice treated neonatally with BPA, severe pathologies of the uterus including adenomyosis, leiomyomas, atypical hyperplasia, stromal polyps were observed [30]. In addition, BPA exposure showed fewer primordial follicles and increased growing follicles [31]. These severe pathological conditions are associated with an increased proliferation rate likely mediated by an estrogenic pathway [31,32]." was added.

Other minor modifications are indicated in the revision.